# Determinants of knowledge on sexually transmitted infections among students in public higher education institutions in Melaka state, Malaysia

**Norain Mansor[1,2], Norliza Ahmad[1]\*, Hejar Abdul Rahman[1]**

**1** Community Health Department, Faculty of Medicine and Health Sciences, Universiti Putra Malaysia, Selangor, Malaysia, **2** Ministry of Health Malaysia, Putrajaya, Malaysia

⊙ These authors contributed equally to this work.

\* lizaahmad@upm.edu.my

## Abstract

### Introduction

The increasing trend of sexually transmitted infections (STIs) among the young population is a significant public health problem. This study aimed to determine the level of knowledge on STIs among students in higher education institutions and its predicting factors, in Melaka.

### Methodology

A cross-sectional study was conducted among 600 students from higher education institutions in Melaka aged between 18 to 30 years old. Multistage sampling of the institutions was performed. Valid and reliable self-administered questionnaire in the national language, Bahasa Malaysia, was used as to collect data on sociodemographic, personal background, knowledge on STIs and sources of information for STIs. Univariate, bivariate and multivariate analyses were conducted using IBM SPSS software version 25.

### Results

The response rate for this study was 88%. The mean knowledge score was 24.1 ±5.1 out of 38. HIV was the most known STIs while gonorrhoea, trichomoniasis and chlamydial infections were among the least known STIs. Oral intercourse was the least known sexual activity that could transmit STIs. Higher proportion of respondents had correct knowledge on control and preventive measures of STIs (between 78% and 95%) compared to correct knowledge on sign and symptoms of STIs (between 8.5% and 67.8%). More than 90% of the respondents were unaware that a person infected with STIs could be symptom free. Four variables were identified as the determinants of the knowledge on STIs, which were level of education, place of stay, history of sexual and reproductive health education and involvement in STIs awareness programs ($F_{(4,445)} = 11.405$, $p < 0.001$, $R^2 = 0.093$).

**Data Availability Statement:** All relevant data are within the manuscript and its Supporting Information files.

**Funding:** The author(s) received no specific funding for this work.

**Competing interests:** The authors have declared that no competing interests exist.

## Conclusions

The knowledge on STIs among students in higher education institutions was unsatisfactory. The existing sexual education programs can be strengthened by delivering more information on other STIs rather than focusing on HIV only. The future program should focus on students of diploma and/or skill certificate and staying off-campus.

## Introduction

Increasing trend of sexually transmitted infections (STIs) among young adults is an alarming issue as they are among the major contributors for STIs cases worldwide [1,2]. The World Health Organization (WHO) has been performing global estimation for four curable STIs approximately every five years since 1995 [3]. The latest report published in June 2019 estimated about 380 million new cases of curable STIs in year 2016, with the WHO Region of the Americas having the highest incidence rate for syphilis and chlamydia among both men and women, while the WHO African region had the highest incidence rate for gonorrhoea and trichomoniasis in both men and women [3]. In Malaysia, the available surveillance systems for STIs reported an increasing trend of syphilis and gonorrhoea cases, with the incidence rate of syphilis being only 5.7 per 100,000 population in 2012 but had increased to 8.0 per 100,000 population in 2017 [4,5]. Similarly, the incidence rate of gonorrhea was 4.78 per 100,000 population in 2013 but had markedly increased to 10.39 in 2017 [5,6]. Nonetheless this marked increase in the incidence of syphilis and gonorrhea might have been contributed by the improvement in the surveillance systems for STIs in the country and the establishment of STI Friendly Clinics in selected government health clinics starting from the year 2015 [7].

Building social relationship, expanding the social network, and searching for romantic partner for stable relationship are part of the normal process in psychosexual development of young adult [8]. At the same time, most young adults are exposed to conducive environment that are favourable for them to engage in risky sexual behaviour based on peer and digital culture influences. The complex interactions between these biological, behavioural and sociocultural factors make them more vulnerable to get STIs compared to the older population [1]. The profound impacts of STIs are not merely the increased risk of HIV infection but also serious implications on the sexual and reproductive health, such as congenital syphilis, cervical cancer and infertility [9,10]. The provision of comprehensive health information, education and health promotion programs are important measures that should be prioritized by each country to end the epidemic of STIs [9]. As certain types of STIs do not present with obvious signs and symptoms, the public should have the knowledge on the different types of STIs and the spectrum of diseases caused by them. In addition, the information regarding the risk factors for STIs is also vital for the implementation of preventive measures [11].

Knowledge on STIs among young adults including undergraduate students in Malaysia is still lacking. For example, approximately 90% of the respondents in one higher education institution in the state of Selangor believed that STIs could be transmitted through handshakes [12]. Another study in several higher education institutions in Negeri Sembilan and Selangor conducted among students in the field of health, reported that only 63.9% of the respondents had knowledge that syphilis is a STI, 45.4% knew gonorrhoea is a STI and only 50% of the respondents knew STIs could be asymptomatic [13]. Additionally, a study conducted among male respondents aged 15 to 24 years in five states in Malaysia reported only 78.7% of the respondents knew that condom is one of the preventive measures of STIs [14].

In Malaysia, various platforms have been used to deliver sexual health information to young adults through school-based syllabus and community programs [15,16]. Under the National Policy in Reproductive Health and Social Education or also known as the PEKERTI policy, most of the programs target adolescents at school through sexual and reproductive health (SRH) topics which were embedded in other subjects such as biology and family health, Islamic education, moral physical and health education subjects.[15–17] Moreover, under the National Strategic Plan Ending AIDS (NSPEA) 2016–2030, all higher education institutions are required to have one compulsory session for education on sexual and reproductive health focusing on STIs, for newly registered students [15]. Furthermore, Malaysia is committed in making Melaka City the nation's representative in the Association of South East Asian Nations (ASEAN) pilot project, known as 'Getting to Zero City' which began in the year 2013 [7,18,19]. "Getting to Zero City" is a regional project of the ASEAN leaders who commit to achieve zero new HIV infections, zero discrimination and zero AIDS-related deaths, in line with global response to end the HIV epidemic by 2030 [18,19]. The projects aim to intensify the efforts in eliminating HIV/AIDS by involving various government agencies and non-profit organizations in activities to key populations and young people. Examples of the activities include screening and harm reduction programs, health education, counselling and treatment services, [19]. Additionally, in response to the NSPEA, the project also has enhanced the existing STIs programs by offering more support and resources for preventive measures such as condom provision, health promotion and health education through STI-Friendly Clinics and various programs at community and organizational levels [7,18,19]. Despite these ambitious initiatives, the level of knowledge on STIs among the young adults, especially in Melaka, is not known.

Therefore, this study aimed to evaluate the knowledge on STIs among undergraduate students in higher education institutions in Melaka, and its predicting factors. The findings of this study can contribute to the efforts towards achieving the objectives of the "Getting to Zero City" project, by identifying the gaps in current educational programs on STIs.

## Methodology

### Study design and population

A cross -sectional study was conducted in the state of Melaka, Malaysia from September 2018 until July 2019. The study population was students from public higher education institutions in Melaka. The selected institutions offered different levels of education programs ranging from l certificate, diploma, and degree levels. The inclusion criteria of study participants were full-time students and aged between 18 and 30 years. The exclusion criteria were those on medical leave, absent during data collection, post-graduate students, and international students. The estimated sample size required to achieve a significance level of 0.05 at 95% confidence interval for the study was 600 based on the mean knowledge score from a previous study [20] after considering the design effect of 1.5 [21] and 20% non-response rate.

The sampling method was multistage sampling. During the first stage, 10 out of 21 public higher education institutions in Melaka were randomly selected using a random number generator. After being granted approval to conduct the study in these 10 institutions, the second stage of sampling was conducted by randomly selecting the study participants from the registration list using a computer-generated random sequence software. The sample size required was allocated proportionately to the number of students in each selected institution. The institutions with a larger number of students contributed to larger samples. The Student's Welfare Department and lecturers of these institutions then informed the students regarding the research to avoid any mistrust among the students regarding the execution of the research. The participants were then contacted by the researcher via phone calls and invited to attend

briefing sessions at the predetermined venue (classroom/hall). During the briefing sessions, the participants were informed about the research and were invited to voluntary participate in it. Only those who volunteered were given the consent form to be signed. Participants who returned the signed consent forms were then given the study questionnaire and were requested to complete the questionnaire before leaving the briefing venue. The researcher was always present to answer any queries.

## Measures

A validated and reliable self-administered questionnaire used in the study was adapted from several studies [12,13,22,23]. The questionnaire consisted of three parts. Part 1 comprised of 13 items related to sociodemographic characteristics and personal background of the respondents. Part 2 was on knowledge of STIs which consisted of five constructs. The first construct consisted of 10 items on type of STIs, the second construct consisted of seven items on the mode of transmission of STIs, the third construct consisted of seven items on the risk factors of STIs, the fourth construct consisted of eight items on the sign and symptoms of STIs, and the fifth construct consisted of six items on the control and preventive measures of STIs. Each item had three-answer options which were 'yes', 'no', or 'don't know'. One score was given to a correct answer and zero score was given to an incorrect answer. The maximum score was 38 and the minimum score was 0. Part 3 comprised of one question on the source of information for STIs. The entire questionnaire was developed in English. Forward and backward translation was performed by an English teacher who holds a master's degree in Art (English Language). Face validity was tested among students from one of the public higher education institutions in Melaka aged 18 to 30 years old who were not included as participants in this study. Amendments were made based on the feedback given by the students. Content validity was performed by three public health physicians. The modifications on the questionnaire were made based on the recommendations given by them.

The questionnaire was tested for its reliability on 60 students, which was 10% of the total sample size. The students have been selected from one of the public higher institutions in Melaka, and they were not included as study participants. The internal consistency of the questionnaires was estimated by using Cronbach's alpha where the value of more than 0.7 was considered acceptable [24]. The value of Cronbach's alpha for each construct was summarized in "Table 1". All items in each construct were retained and all constructs were retained to give the overall Cronbach's alpha of 0.83 for knowledge of STIs.

## Variables

The dependent variable in this study was knowledge scores on STIs while independent variables were sociodemographic factors (age, sex, marital status, ethnicity, religion and level of

**Table 1. Summary of the Cronbach's alpha index value.**

| Construct | Number of items | Mean (SD) | Cronbach's alpha index value |
|---|---|---|---|
| Knowledge of type of STIs | 10 | 4.13 (2.07) | 0.69 |
| Knowledge on mode of transmission of STIs | 7 | 3.90(2.21) | 0.79 |
| Knowledge on risk factors of STIs | 7 | 4.57(2.05) | 0.77 |
| Knowledge on sign and symptom of STIs | 8 | 3.70(2.43) | 0.79 |
| Knowledge on control and prevention of STIs | 6 | 4.53(2.19) | 0.92 |

Note: SD: Standard deviation.

education), personal background (type of academic course, place of stay, history of sexual and reproductive health education at school, involvement in STIs program and parent's education level). The "level of education" refers to the education levels of respondents at the point of data collection whether the respondents currently taking Malaysian Skill Certificate, diploma, or a bachelor's degree. Those who were studying for a degree, they will be classified as "degree" and those who were studying for a diploma and/or skill certificate were classified as "non-degree" respondents. For the "type of academic course", courses in fashion, art and design, photography, creative imaging, hospitality management, finance management, business administration, and education were considered as "art courses", while "science courses" refer to courses related to engineering namely technology communication engineering, computer engineering, industrial and manufacturing engineering, mechanical engineering, automotive, electrical engineering. "Involvement in STIs program" refers to the involvement of respondents in any programs or health talks on STIs organized in their schools, college, or in the community. The type of program referred to the latest STIs program the respondents participated in, be it programs organized in the schools, college, or community settings. History of sexual and reproductive health education at school was asked to ensure the participants were aware regarding SRH education delivered in the schools since there were no specific curriculum or subject named as SRH in school because SRH topics were embedded in other subjects such as biology and family health, Islamic education, moral physical and, health education subjects [17].

## Statistical analysis

The data were described and analyzed using the International Business Machines Statistical Package for Social Sciences (SPSS) version 25.0. Normality testing was performed on continuous variables which included the total knowledge scores and the age of the respondents.

Descriptive statistic or univariate analysis was used to assess the frequency distribution, the mean and standard deviation for knowledge score of STIs by sociodemographic and personal backgrounds. The bivariate analysis used in this study was independent t-test except for three variables that did not fulfil the assumptions for independent t-test which were ethnicity, religion, and marital status. The data for the knowledge score for each group in these three variables were not normally distributed. Therefore, Mann-Whitney U test was used in bivariate analysis for ethnicity, religion, and marital status. Multiple linear regression (MLR) was used to identify the determinants for the knowledge score of STIs. Covariates (level of education, place of stay, history of SRH, and involvement in STIs program) were also adjusted using MLR. Variables with a significance level of 0.25 were chosen to be imputed into the model [25]. Dummy variables were created and coded as 0 and 1, where 0 was labelled as the reference group. Stepwise selection approach was used, as it provided the most parsimonious model. Assumptions of normally distributed residuals were fulfilled for total knowledge scores using graphical methods. Significance was predetermined at a probability value of 0.05 and less.

## Ethical approval

This research project was approved by the Ethics Committee for Research Involving Human Subjects Universiti Putra Malaysia (UPM/TNCPI/RMC/1.4.18.2 (JKEUPM)). This study was also granted permission by the directors and Vice Chancellors of each institution. Written consent was obtained from all participants. Participants were informed that their involvement in this research was voluntary and that all information they provided remain confidential. All the data and documents from this study will be kept in locked storage and will be disposed of according to the standard operating procedure, five years after the completion of the study.

## Results

The total number of respondents approached by the researcher was 700. Of this, 680 students were found to be eligible. Another 20 students were not eligible due to various reasons such as being on the semester break and doing practical training outside the campus. A total of 680 self- administered questionnaires were disseminated to these eligible respondents and all of the questionnaires were returned to the researcher within the allocated time. However, 80 questionnaires were incomplete (more than 5%) hence were excluded during the final analysis. Therefore, the response rate was 88.23%.

Majority of the respondents were of Malay ethnicity (95.8%), single (99.3%) and aged between 19 and 21 years (61%). The mean age of the respondents was 21 ± 1.59 years. The distribution of male and female respondents was almost equal. "Table 2" shows the respondents' sociodemographic characteristics.

The overall mean knowledge score was 24.09 ±5.06 out of 38. "Table 3" shows respondents' replies on the knowledge of STIs. The most known STI was HIV while the least known STIs were trichomoniasis (12.5%), chlamydia (14.3%), and gonorrhoea (19.0%). Majority of the respondents were aware of the mode of transmission of STIs, which was through vaginal intercourse (90%) and anal intercourse (80%). However, on average, about one-third of respondents provided incorrect answe14rs especially on vertical transmission (39%), blood transfusion (37%), and oral intercourse (36%). In terms of risk factors of STIs, the top three risk factors that were misunderstood by the respondents were uncircumcised male (76%), smoking (33%), and early sexual debut (26%). About 90% of the participants did not know that those who are infected with STIs may not have any signs and symptoms. The top three control and preventive measures of STIs that the respondents were unaware of are: being faithful in an intimate relationship (23%), using condoms during sexual activity (15%), and vaccination against certain types of STIs (14%).

Most of the respondents obtained information on STIs from three major sources which were the internet (41.3%), lessons at school or college (27.8%) and health personnel (16.7%). Parents and friends contributed to only 1% and 2% as source of information on STIs respectively. Other sources of information regarding STIs are as tabulated in "Table 4".

From both independent t-test and Mann-Whitney U test, six variables were found to be significantly associated with the level of knowledge on STI, with p-value 0.05 and less. These variables were age, education level, type of course, history of SRH lesson, involvement in STIs program, and place of stay, as shown in "Table 5".

The variables with p-value of 0.25 and less were chosen to be imputed into the preliminary model in the multiple regression analysis. These variables were age, marital status, level of education, type of course, history of SRH lesson, involvement in STIs program, type of undergraduate program, and place of stay. In the final model, four variables were identified to be statistically significant in predicting the level of knowledge [F (4,445) = 11.405, p <0.001, R2 = 0.093]. The significant variables were level of education, history of SRH education, involvement in STIs program, and place of stay as shown in "Table 6". Respondents' predicted knowledge scores were equal to 25.06 + 1.89 (level of education)– 2.03 (place of stay)– 1.18 (history of SRH education)– 1.08 (involvement in STI program). Respondents who were studying in the degree program scored 1.89 more than those who were not, those staying out of campus scored 2.03 less than those staying in campus, those without a history of SRH education scored 1.18 less than those with history of SRH education, and those who were not involved in STIs program scored 1.08 less than those who were. These four variables explained 9.3% of the variation in the knowledge score of STIs, and each variable explained 8.5% variation in the knowledge score of STIs while another 91.5% variance in the knowledge score of STIs was explained by other factors which were not studied.

**Table 2. Sociodemographic characteristics and personal background of respondents.**

| Characteristics | n | (%) |
|---|---|---|
| **Age (Years)** | | |
| <23 | 482 | 80.3 |
| ≥23 | 118 | 19.7 |
| **Sex** | | |
| Male | 304 | 50.7 |
| Female | 296 | 49.3 |
| **Ethnicity** | | |
| Malay | 575 | 95.8 |
| Chinese | 8 | 1.3 |
| Indian | 4 | 0.7 |
| Others | 13 | 2.2 |
| **Marital status** | | |
| Single | 596 | 99.3 |
| Married | 4 | 0.7 |
| **Religion** | | |
| Muslim | 581 | 96.8 |
| Buddha | 7 | 1.2 |
| Hindu | 4 | 0.7 |
| Christian | 8 | 1.3 |
| **Education level** | | |
| Skill Certificate | 167 | 27.8 |
| Diploma | 178 | 29.7 |
| Bachelor | 255 | 42.5 |
| **Type of course** | | |
| Art | 323 | 53.8 |
| Sciences | 277 | 46.2 |
| **Place of stay** | | |
| In campus/hostel | 377 | 62.8 |
| Out campus | 223 | 37.2 |
| **History of sexual and reproductive health education at school** | | |
| Yes | 307 | 51.2 |
| No | 143 | 23.8 |
| Unable to recall | 150 | 25.0 |
| **Involvement in STIs program** | | |
| Yes | 328 | 54.7 |
| School program | 106 | 32.3 |
| Not school program | 222 | 67.7 |
| No | 272 | 45.3 |
| **Father's education** | | |
| No formal education | 10 | 1.7 |
| Primary school | 43 | 7.2 |
| Secondary school | 285 | 47.5 |
| College/university | 262 | 43.7 |
| **Mother's education** | | |
| No formal education | 14 | 2.3 |
| Primary school | 35 | 5.8 |
| Secondary school | 330 | 55.0 |
| College/university | 221 | 36.8 |

Note: STI: Sexually transmitted infection.

## Discussion

Sexually transmitted infections among young adults are increasing worldwide including Malaysia [1,3,5,9]. Policy and programs have been put in place [11,16] in order to manage these trends. This cross-sectional study aimed to determine the level of knowledge on STIs and its determinants among undergraduate students of higher education institutions in Melaka. This study was conducted 5 years after the implementation of the "Getting to Zero City" pilot project. The present study yielded a good response rate that was contributed by the commitment and cooperation of the institutions' management including lecturers and students' affairs departments. Majority of the respondents were Malay, and this could be due to the Malay ethnicity being the largest ethnic group in Melaka and majority of the public higher education institutions in Melaka is accommodated by Malay ethnics [26].

The knowledge of STIs among the respondents in this study was slightly higher than other studies conducted among students in higher institutions and the general population. In the present study, the mean knowledge score was 24 out of 38 scores (63%) compared to another study conducted among university students in the central zone of Malaysia (53%) [13] and university students in Thailand (49%) [27]. While a study conducted among patients who attended a venereal clinic in Malaysia had reported the mean knowledge score of 12 out of 33 (36%) [23]. The higher knowledge scores observed in our study might be contributed by the recent information received by the participants since all higher education institutions must organize an awareness program on HIV/STI once a year in response to the NSPEA 2016–2030 [11]. As expected, HIV was the most well-known STIs among the respondents. This finding was consistent with findings from previous studies among the young population [12–14,22,27–31]. Lesser awareness on other STIs was also observed among young people in previous studies conducted locally [12–14]. This could be due to lesser attention given to the other STIs during health talks or programs.

Generally, the respondents knew that STIs were not only transmitted through sexual activities but also through other routes such as vertical transmission, blood transfusion, and oral intercourse. Similar findings were reported by previous studies conducted among higher institution students in Malaysia and Saudi Arabia [12,13,28]. Amongst the three common sexual activities asked in the present study, oral intercourse was the least known sexual activity that can act as a mode of transmission for STIs. Oral sex is being practiced by both heterosexual and homosexual groups and usually perceived as low risk or safer activity [32]. However, our finding was not consistent with a study conducted among the first-year undergraduate in one of the universities in Turkey, where more respondents knew about oral intercourse as a mode of transmission of STIs as compared to vaginal intercourse [22]. Our study showed that less than one-third of the respondents knew that being uncircumcised (for males) is one of the risk factors for STIs. This finding is inconsistent with a study conducted among males who attended the clinic for HIV testing in Namibia where 66% of the respondents knew male circumcision could reduce STIs transmission [33]. Furthermore, one-third of our respondents thought smoking is one of the risk factors for STIs. They might be confused with the indirect health effects of smoking that could lead to an immunocompromised state and could lead the individual to contract infectious diseases [34].

Among all the components of knowledge on STI, our respondents had the poorest level of knowledge on the signs and symptoms of STIs Our findings are consistent with previous studies [12–14,22,27–29]. More than 90% of the respondents were unaware that a person infected with STIs could be showing no sign or symptoms. These findings could explain the late detection of STIs cases among both males and females [1–3,9]. In addition, half of the respondents thought that a prolonged cough of more than two weeks is a sign and symptom for STIs. Since

**Table 3. Knowledge of STIs.**

| Knowledge of STIs | Correct Answer | | Incorrect answer | |
|---|---|---|---|---|
| | n | % | n | % |
| **Type of STIs** | | | | |
| Syphilis | 327 | 54.5 | 273 | 45.5 |
| Pneumonia | 460 | 76.7 | 140 | 23.3 |
| HIV | 554 | 92.3 | 46 | 7.7 |
| Tuberculosis | 426 | 71.0 | 174 | 29.0 |
| Trichomoniasis | 75 | 12.5 | 525 | 87.5 |
| Genital herpes | 209 | 34.9 | 391 | 65.1 |
| Hepatitis B | 149 | 24.8 | 451 | 75.2 |
| Gonorrhoea | 114 | 19.0 | 486 | 81.0 |
| Chlamydia | 86 | 14.3 | 514 | 85.7 |
| Measles | 345 | 57.5 | 255 | 42.5 |
| **Mode of transmission** | | | | |
| Vaginal intercourse | 537 | 89.5 | 63 | 10.5 |
| Anal intercourse | 479 | 79.8 | 121 | 20.2 |
| Oral intercourse | 384 | 64.0 | 216 | 36.0 |
| Using public toilet | 414 | 69.0 | 186 | 31.0 |
| Blood transfusion | 377 | 62.8 | 223 | 37.2 |
| Vertical transmission | 365 | 60.8 | 235 | 39.2 |
| Hand shaking | 438 | 73.0 | 162 | 27.0 |
| **Risk factors** | | | | |
| Sexual activity without condom | 518 | 86.3 | 82 | 13.7 |
| Multiple sexual partner | 572 | 95.3 | 28 | 4.7 |
| Same sex relationship | 487 | 81.2 | 113 | 18.8 |
| Having sex with sexual worker | 552 | 92.0 | 48 | 8.0 |
| Early sexual debut | 444 | 74.0 | 156 | 26.0 |
| Uncircumcised male | 147 | 24.5 | 453 | 75.5 |
| Smoking | 401 | 66.8 | 199 | 33.2 |
| **Sign and symptoms** | | | | |
| Itchiness at genitalia | 386 | 64.4 | 214 | 35.6 |
| Abnormal vaginal or urethral discharges | 407 | 67.8 | 193 | 32.2 |
| Vomiting | 286 | 47.7 | 314 | 52.3 |
| Pain during sexual intercourse | 393 | 65.5 | 207 | 34.5 |
| Ulcer at genitalia | 335 | 55.8 | 265 | 44.2 |
| Headache | 312 | 52.0 | 288 | 48.0 |
| Prolonged cough more than 2 weeks | 297 | 49.5 | 303 | 50.5 |
| Showing no sign and symptom | 51 | 8.5 | 549 | 91.5 |
| **Control and prevention** | | | | |
| Using condom during sexual activity | 513 | 85.5 | 87 | 14.5 |
| Avoid premarital sexual intercourse | 553 | 92.2 | 47 | 7.8 |
| Be faithful in the intimate relationship | 465 | 77.5 | 135 | 22.5 |
| Do screening test | 556 | 92.7 | 44 | 7.3 |
| Get early treatment | 570 | 95.0 | 30 | 5.0 |
| Get vaccinated to protect against certain type of STIs | 514 | 85.7 | 86 | 14.3 |

Note: STI: Sexually transmitted infection.

**Table 4. Distribution of sources of information for STIs among respondent (N = 600).**

| Source of information | n | % |
|---|---|---|
| Lesson at school/college | 167 | 27.8 |
| Internet | 248 | 41.3 |
| Television | 13 | 2.2 |
| Radio | 4 | 0.7 |
| Reading materials | 40 | 6.7 |
| Parent | 8 | 1.3 |
| Friends | 13 | 2.2 |
| Partner | 5 | 0.8 |
| Health personnel | 100 | 16.7 |
| Others | 2 | 0.3 |

Note: STI: Sexually transmitted infection.

tuberculosis is one of the commonest co-infections among HIV patients [35,36] the sign and symptoms of tuberculosis may be thought to be associated with STIs.

In general, our findings show that the respondents had adequate knowledge on the control and preventive measures of STIs compared with several previous studies conducted locally among the young population in Malaysia [12–14,23]. This could be contributed by the HIV/STI education program that has been carried out. In addition, the mandatory requirement of HIV screening tests before marriage for Muslim couples in Malaysia might have resulted in increased awareness on the importance of avoiding pre-marital sexual activity and getting screening tests done [37]. However, we expected the proportion of the respondents who were aware of the role of the condom as a preventive measure for STIs would be higher than that found in this study which was 85%. Furthermore, about 85% of our respondents were aware of the role of vaccination in protecting against STIs, which was higher than the findings in one of the studies conducted among university students at Taif, Saudi Arabia. Their findings showed that only 38% of their respondents knew the existence of vaccination as one of the protective methods against STIs [28]. Our findings could be contributed by the free vaccination services for two viral STIs namely Hepatitis B and Human Papillomavirus (HPV) for Malaysians. The existence of this free vaccination could contribute to the increase in the awareness on the availability of vaccines to protect against certain STIs.

The present study found that the internet is the main source of information for STIs followed by the syllabus at school or college and health personnel. This could be attributed by the high percentage of the young population who are smartphone users in Malaysia. The Malaysian Communications and Multimedia Commission (MCMC) reported that 88% of young people aged between 20 and 30 years are smartphone users with 30% of them are internet users in year 2018 [38,39]. Furthermore, a study showed that about half of polytechnic students in Malaysia spend their leisure time surfing on the internet [40]. These data could explain why the internet was the most preferred channel to gain information for STIs in our study. Our findings are inconsistent with previous studies conducted in other countries. For example, the main source of information among university students in Turkey was reading materials such as books and magazines [22] and in Thailand was lesson at school/college [27]. In Nigeria, radio and television were the main sources for STIs information among secondary schools' students [29], while in another study conducted among youth in Italy, parents and teachers were the main sources of information for STIs [30].

**Table 5. Association between sociodemographic and personal factors and knowledge score of STIs.**

| Variables | N | Knowledge score of STIs | Test statistic t/Z | p- value |
|---|---|---|---|---|
| Age (years) | | | | |
| <23 | 482 | 23.82 (5.198) [c] | -1.365a | 0.003* |
| ≥23 | 118 | 25.19 (4.291) [c] | | |
| Sex | | | | |
| Male | 304 | 23.97 (5.293) [c] | -0.587a | 0.557 |
| Female | 296 | | 24.21 (4.809) [c] | |
| Ethnicity | | | | |
| Malay | 578 | 24.00 (7.000) [d] | -0.914b | 0.361 |
| Non- Malay | 22 | | 24.00 (5.000) [d] | |
| Religion | | | | |
| Muslim | 582 | 24.00 (7.000) [d] | -0.100b | 0.921 |
| Non-Muslim | 18 | | 26.00 (5.000) [d] | |
| Marital status | | | | |
| Not married | 596 | 24.00 (7.000) [d] | -1.522b | 0.128 |
| Married | 4 | | 28.50 (9.000) [d] | |
| Education level | | | | |
| Not degree | 345 | 23.52 (5.379) [c] | -3.296a | 0.001* |
| Degree | 255 | | 24.85 (4.488) [c] | |
| Type of course | | | | |
| Art | 323 | 24.52 (4.669) | 2.230 | 0.026* |
| Science | 277 | 23.59 (5.444) | | |
| History of SRH lesson | | | | |
| Yes | 307 | 24.84 (4.746) | 3.454 | 0.001* |
| No | 143 | 23.01 (5.459) | | |
| Involvement in STIs program | | | | |
| Yes | 328 | 24.57 (5.007) | 2.581 | 0.010* |
| No | 272 | 23.51 (5.069) | | |
| Type of program | | | | |
| School program | 106 | 25.08 (4.663) | -1.318 | 0.189 |
| Not school program | 222 | 24.32 (5.157) | | |
| Place of stay | | | | |
| In campus | 377 | 24.63 (5.014) | 3.439 | 0.001* |
| Out campus | 223 | 23.17 (5.013) | | |
| Father's education | | | | |
| Not going to college | 338 | 24.09 (5.036) | 0.009 | 0.992 |
| Going to college | 262 | 24.09 (5.098) | | |
| Mother's education | | | | |
| Not going to college | 379 | 24.25 (4.958) | 1.036 | 0.301 |
| Going to college | 221 | 23.81 (5.226) | | |

Note:

*Significant at p ≤ 0.05,

[a] Independent t-test,

[b] Mann-Whitney U test,

[c] Mean (Standard Deviation),

[d] Median (Interquartile Range).

STI: Sexually transmitted infection; SRH: Sexual and reproductive health.

**Table 6. Determinants of knowledge score on STIs.**

| Variable | Unstandardized coefficient | | Standard coefficient | t | p-value | 95% CI | |
|---|---|---|---|---|---|---|---|
| | B | Std error | Beta | | | Lower bound | Upper bound |
| Constant | 25.060 | 0.387 | | 64.752 | <0.0001 | 24.300 | 25.821 |
| Level of education | 1.889 | 0.485 | 0.185 | 3.898 | <0.0001* | 0.937 | 2.841 |
| 0 = Not degree** | | | | | | | |
| 1 = Degree | | | | | | | |
| Place of stay | -2.028 | 0.478 | -0.194 | -4.241 | <0.0001* | -2.968 | -0.170 |
| 0 = In campus** | | | | | | | |
| 1 = Out campus | | | | | | | |
| History of SRH education | -1.178 | 0.513 | -0.109 | -2.297 | 0.0220* | -2.187 | -0.489 |
| 0 = Yes** | | | | | | | |
| 1 = No | | | | | | | |
| Involvement in STIs program | -1.078 | 0.491 | -0.106 | -2.194 | 0.029* | -2.044 | -1.088 |
| 0 = Yes** | | | | | | | |
| 1 = No | | | | | | | |

** Reference group,

*Significant at $p < 0.05$, $R^2 = 0.093$, Adjusted $R^2 = 0.085$.

STI: Sexually transmitted infection; SRH: Sexual and reproductive health.

About one third of our respondents claimed to have received information on STIs through lessons given in schools or colleges. Even though sexual and reproductive health was taught in school, a study showed that the information delivered was perceived as inadequate [41]. A study conducted among 1706 university students on what they had received during school-based sexual education found that information on STIs was lacking [42]. In addition, a study concluded that cooperation between schools, young people, families, and communities is vital to amplify the success of sexuality education [43]. However, our findings showed that parents and friends were the least relied on as source of information on STIs. Although parents are expected to provide sexual education to their offspring, usually they are reluctant to do so as they assume that will indirectly provide permission for their children to explore on sexual behaviors [44]. Our findings were in contrast with a study conducted among male university students in Saudi Arabia where friends were the main source of information for STIs and were reported to be a significant factor in determining the knowledge of STIs [28]. A meta-analysis showed that peer education approaches have significantly contributed to increase knowledge on HIV (OR: 0.37; 95% CI: 1.88, 2.75) [45].

From regression analysis, four determinants were found to be significantly associated with knowledge of STIs in this study. These factors were level of education, place of stay, history of SRH education and involvement in STIs programs. In the present study, the respondents with higher education level were found to have better knowledge of STIs. Similar result was reported by previous studies [21,46]. A study conducted in Iran among women aged 15 to 49 years showed that women whose education level was lesser than the degree level had nearly 5 times lower knowledge score than those who have a degree [21]. A study conducted among high school students in the United States showed that the students in higher grade had better knowledge score than those in the lower grade [46]. History of receiving SRH education at school was found to be a significant factor influencing the knowledge of STIs in our study. Therefore, Malaysia should continue and improve the implementation of SRH education in schools [42] as well as at tertiary levels organized by government and non-governmental organizations [15,16].

Another significant factor for knowledge of STIs found in our study was place of stay. The respondents who lived on campus had significantly better knowledge of STIs as opposed to those who lived out of campus. Under the National Strategic Plan Ending AIDS 2016–2030, all higher educational institution must organize an educational program on HIV/STIs once a year [15]. In addition, on campus students need to adhere to their institution's regulation of compulsory attendance of organized programs in order to obtain merit for continuity of stay on campus. Therefore, students who lived on campus might have recently exposed to the information on HIV/STIs and these influenced their knowledge on STIs.

The strength of this study includes the random sampling method used during the selection of study's participants. Additionally, students from half of the public higher education institutions available in Melaka were included in this study, which can justify the generalizability of this study's findings among the study population. Melaka state which was chosen to represent Malaysia in the "Getting to Zero City" was a suitable location to assess the knowledge on HIV and other STIs infections. Moreover, a validated and reliable questionnaire used in this study was tailored and modified to suit our local context, thus the information bias can be further minimized. In terms of study limitation, the findings from this study could not represent students in higher education institutions in Malaysia because the study populations were restricted to one state only which was Melaka state. Besides, several other factors which could influence the knowledge of STIs were not examined in this study such as sexual practices and history of STIs. Selection bias might have occurred during the selection of the study's participants since they might feel obliged to join the study because they were instructed by the college administrators. However, the researchers have attempted to reduce the selection bias by randomly selecting the participants from the list of registered students and personally approach them by phone calls and offered voluntarily participation. Another possible bias that could have occurred in this study is information bias. Some respondents may be hesitant in answering the questions truthfully due to the sensitive and personal nature of the topic being asked in our local setting. Besides, recall bias could occur on events such as exposure to SRH education and involvement in STIs programs. However, the researcher has attempted to reduce the bias by ensuring the validity and reliability of the questionnaires.

## Conclusion

The current findings show that the knowledge level on STIs has slightly increased compared to those in previous local studies, but it was still unsatisfactory. Knowledge on STIs was mainly obtained from the internet and the main STI known was HIV.

As previous exposure to SRH education and involvement in STIs programs are the determinants for knowledge on STIs, the existing education programs in the country should be continued and enhanced, by conveying more information on other types of STIs rather than focused solely on HIV. Besides, the target group of future programs should focus on students of diploma and/or skill certificates and staying off-campus. Future research also should focus on studying other factors that can contribute to the low level of knowledge on STIs.

## Supporting information

**S1 Appendix. Questionnaires.**
(XLSX)

## Acknowledgments

The authors would like to express their gratitude towards HIV/STI Unit of Melaka's Health Department for sharing valuable data on STIs. We are very thankful for the cooperation and

commitment given by all the higher education institutions and undergraduate students involved in this study. The authors also would like to thank the Director General of Health Malaysia for his permission to publish this article.

## Author Contributions

**Conceptualization:** Norain Mansor, Norliza Ahmad, Hejar Abdul Rahman.

**Formal analysis:** Norain Mansor, Norliza Ahmad.

**Investigation:** Norain Mansor.

**Methodology:** Norain Mansor, Hejar Abdul Rahman.

**Supervision:** Norliza Ahmad, Hejar Abdul Rahman.

**Validation:** Hejar Abdul Rahman.

**Visualization:** Norliza Ahmad.

**Writing – original draft:** Norain Mansor.

**Writing – review & editing:** Norliza Ahmad.

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
