## [Decision Letter · Decision Letter 0]

28 Jun 2020

PONE-D-20-11879

Knowledge on sexually transmitted infections among undergraduate students in “Getting to Zero City”, Malaysia: A cross-sectional study

PLOS ONE

Dear Dr. Ahmad,

Thank you for submitting your manuscript to PLOS ONE. After careful consideration, we feel that it has merit but does not fully meet PLOS ONE’s publication criteria as it currently stands. Therefore, we invite you to submit a revised version of the manuscript that addresses the points raised during the review process.

We look forward to receiving your revised manuscript.

Kind regards,

Siyan Yi, MD, MHSc, PhD

Academic Editor

PLOS ONE

2. Please address the following:

- Please include additional information regarding the survey or questionnaire used in the study and ensure that you have provided sufficient details that others could replicate the analyses. For instance, if you developed a questionnaire as part of this study and it is not under a copyright more restrictive than CC-BY, please include a copy, in both the original language and English, as Supporting Information.

- Please ensure you have thoroughly discussed any potential limitations of this study within the Discussion section.

Reviewers' comments:

Reviewer's Responses to Questions

**Comments to the Author**

1. Is the manuscript technically sound, and do the data support the conclusions?

Reviewer #1: Yes

Reviewer #2: Yes

Reviewer #3: Yes

Reviewer #4: Partly

Reviewer #5: Yes

2. Has the statistical analysis been performed appropriately and rigorously? 

Reviewer #1: Yes

Reviewer #2: Yes

Reviewer #3: Yes

Reviewer #4: Yes

Reviewer #5: Yes

3. Have the authors made all data underlying the findings in their manuscript fully available?

Reviewer #1: Yes

Reviewer #2: Yes

Reviewer #3: Yes

Reviewer #4: No

Reviewer #5: Yes

4. Is the manuscript presented in an intelligible fashion and written in standard English?

Reviewer #1: Yes

Reviewer #2: Yes

Reviewer #3: Yes

Reviewer #4: No

Reviewer #5: Yes

5. Review Comments to the Author

Reviewer #1: There are various issues that authors need to rectify in this paper.

1- Was this study representative of the Malaysian population? In my NO, because your study mainly comprised of the Sample from the Melaka state and on top of that the Majority of the respondents are Muslims, so its really challenging to make it compatible with the diverse Malaysian population. However, if the title and methods and other parts where authors have tried to justify this study as a representative of Malaysian population are restricted to Melaka only then it will be more suitable for this study.

Title must have "Melaka" and "Malay Muslims" because your data is more representative to this population instead of the whole Malaysia.

2- Abstract: Results section, add some more results that give an idea to the readers who were your main study population. Change "Marks" to score.

3- Introduction: some facts from this report should be in the introduction to give a better idea about the current STI situation in Malaysia https://www.moh.gov.my/moh/resources/Penerbitan/Laporan/Umum/Report_GAM_2019_(Final).pdf

it will be ideal if the following papers are cited as well and the results are compared and contract with them

"Analysis on sex education in schools across Malaysia. Johari Talib , Maharam Mamat, Maznah Ibrahim & Zulkifli Mohamad"

https://www.ncbi.nlm.nih.gov/pmc/articles/PMC5334713/

4- Methods, Need more explanation about study respondents? also provide details how multistage sampling was performed and how many institutes were approached for consent to this study. For reliability analysis how many subjects were invited for the pilot testing? were these included in the final analysis or excluded. Share the reliability analysis tables with Scale items if deleted options so that readers can see what was the reliability of each item.

Standardize throughout MARKS to SCORE

Analysis: in the abstract author claim " Univariate, bivariate and multivariate analyses

were conducted using IBM SPSS software version 25" was performed but in the methods information is missing about this part. need to provide details which variables was included for which analysis what were the co-varaites etc

5- RESULTS

- Reliability and validity additional data need to be provided

- Chi-sq was applied which give association among the variables, I suggest to see the difference among the group not the association. go for t-test and ANOVA to achieve this objective

- Results from the regression model need to be explained in detail and its interpretation need to improved in the results and discussion section

6- Limitation section is missing

7- Conclusion need revision after re-analysis of the results

Reviewer #2: Re:

Manuscript #: PONE-D-20-11879

Title: Knowledge on sexually transmitted infections among undergraduate students in “Getting to Zero City”, Malaysia: A cross-sectional study

Authors: norain binti mansor; Norliza Ahmad; Hejar Abdul Rahman

Mansor et al. have presented data on the lack of knowledge on sexually transmitted infections among undergraduate students in Malaysia. Although this is an important theme, the methods are incomplete.

Some of my concerns, questions and suggestions regarding this manuscript are outlined below:

1. Some of the abbreviations were not identified in the context, e.g. AOR.

2. Line 117; Reference linked with the sample size estimation; it is not clear how the reference supports the statement.

3. Line; 150; Is it True that calculated total knowledge score can be considered as continuous variables? Please define how the total knowledge score was calculated.

4. Line 155; Was the constructed logistic regression model adjusted to all variables observed in the study? Please define as appropriate within the context.

5. Table 1; different religions seem not to be represented equally in utilised sample, especially "Buddha"? Any explanation?

6. Table 5; What is the value and role used to dichotomise the total knowledge score into two groups, namely; Adequate and Inadequate? And what is the rationale behind that?

7. Table 2; Revise percentage for " Do screening test", 92.7+7.8=100.5!.

8. Line 339; please state limitation to this study.

BW

Reviewer #3: Given the current increase in the incidence rates for a variety of sexually transmitted infections in the Asia-Pacific region this paper is timely and relevant. The study design is fairly basic but adequate to address the research questions. THere are some issue with how the paper was presented which must be addressed before the paper can be considered for publication.

Introduction

- In Malaysia, the available surveillance systems for STIs reported

an increasing trend of syphilis and gonorrhoea cases, with the incidence rate of syphilis being

only 5.7 per 100,000 population in 2012 but had increased to 8.0 per 100,000 population in

2017 [4, 5]. Similarly, the incidence rate of gonorrhoea was 4.78 per 100,000 population in

2013 but had markedly increased to 10.39 in 2017

These seem to be quite large increases in incidence in malaysia. Is it possible this could also be connected to changes in the surveillance system itself? Is an increase in reporting possible?

- The second sentence of the second paragraph starting ' The interactions between their internal conflicts..." does not make sense. I don't know what this sentence is trying to say

- In the introduction overall, the reader is left a little confused as to what STIs we are interested in. The title and its reference to "Zero city" implies that we will focus on HIV. In the introduction though we mainly hear about syphilis and gonorrhoea. Need to be clear up front what you are interested in analysing.

Methods

- There is a missing step in the methods section between describing the study population and how they were sampled and then the measures section. We need to know how the students were actually recruited and by who. Was the contact through their education institution? Or did the researchers approach them directly? How were the questionnaires actually administered? Online? This research is on quite a sensitive subject. The details of what i have asked for here are important for assessing the results of the research. For example, people may answer an online survey more openly than a face to face interview.

- To interpret Table 2 we need more specifics about what the actual questions asked were, and what the correct answers were. Don't assume your audience know these things already. For example in Table 2 I assume that there was a question like: " which of the following are STIs" but that is never explicitly said in the methods to this paper. Continuing on i assume that there is a question like: "Which of the following are modes of transmission for STIs" but i don't know for sure.

Results

- You are missing a table 4. You go straight from 3 to 5.

- In Table 5 you have categories of knowledge as 'Adequate' or 'in adequate'. I may have missed it but i can't find definitions for these categories. This should be in the Methods somewhere. Even if it is in the Methods you should include it again as a footnote to the table.

Discussion

- You observe that your study sample had a higher knowledge score than was found in other studies. I know this could be for a variety of reasons but in the discussion you should think about what these reasons may be. For example was the Thai study in second para of Discussion also amongst university students?

Reviewer #4: Why getting to zero city included in the title? Overall title needs to be revised.

Rationale needs to be included. What we already know about this title is missing.

How knowledge was measured? What was the overall Knowledge level? Why three options; Yes, No and DN were included. To see the Knowledge, Yes OR NO can be sufficient. Did you validated / pretested the tool in local context?

Objectives and title both should be revised as per the results. Author has analyzed the association and predictors of Knowledge as well.

Conclusion also needs to be revised.

Overall this needs lot of work to be revised.

Reviewer #5: Knowledge on sexually transmitted infections among undergraduate students in

“Getting to Zero City”, Malaysia: A cross-sectional study

Journal: Plos One

Reviewer’s comments

The manuscript by Mansor et al. presents the results of an original study. They collected data about STI knowledge from undergrad students in a large city in Malaysia using questionnaires.

The study conduct, statistic methods use, and other analyses are appropriate.

The reasonable conclusions are presented. To me the language use is understandable.

I have several comments as follows;

Title

-The full title should include the city name “Mekala”

-The short title on the title page

“Predictors of knowledge of sexually transmitted infections among undergraduate students.”

is different from the one in the submission system

“Knowledge on sexually transmitted infections among undergraduate students in Malaysia.”

Please reconcile.

Abstract

-Introduction: the first sentence “….is an alarming issue”, please specify that it is an alarming issue of what.

-Results: The response rate for this study was 88%. Please specify whether the authors mean response rate participate in the study (other 12% denied to join) or questionnaire completion rate (12% of questionnaires were not returned, or 12% of questions in the questionnaires were not answered).

-Conclusion: what did the authors mean by the term “lower level of education” (all were undergrad student; did they mean the first year student? It should be clarified)

Introduction

In the 4th paragraph, please provide background information about the “Getting to zero” project. Was it implemented in the city or in the university campus. Who were the target population that joined the activities? What kind of thing can people learn from the project?

Methodology

-Ethic approval: typo error in the last sentence, it should be “…will be disposed five years after the completion of the study”

Results

-Firstly, the same question as in the abstract: The response rate for this study was 88%. Please specify whether the authors mean response rate participate in the study (other 12% denied to join) or questionnaire completion rate (12% of questionnaires were not returned, or 12% of questions in the questionnaires were not answered).

-Table 1: Does educational level mean the current or the highest level ever finished?

-involvement in STIs program: please clarify in the footnote or somewhere to be referred to what STIs program means.

-Sociodemographic: Did the authors collect data on sexual behaviors of study participants i.e. gender role, sexual experience, age at sexual debut, sexual partners? Those are interesting and useful information among this population.

-The 2nd paragraph: line 8, “the top three risk factors that were answer wrongly”. The term should be modified to “the top three risk factors that were misunderstanding”

-Table 5 why the authors used the cut-off age at 23 years?

Some categories are not understandable to general reader, i.e. school vs. not school program.

In “History of SRH education” there are 2 groups, yes and no.

If it was the standard SRH education in school, why not everyone has ever attended them prior to continue their higher education?

Were there any differences in characteristics of student who stay in and out of campus? Was it more expensive to stay outside? Were there any specific inclusion criteria for students who get accommodation in the campus i.e. come from far away town, study in some faculties, or final year student who need to work late at night? Those might be confounders which create difference between those who live in and out of campus.

Discussion

In the first paragraph, the author mentioned including lecturers and students’ affair department in the study. This would make the study more vulnerable for coercion. Please explain how they involved in the study recruitment or other activities.

What were the interventions in the “Getting to zero city” pilot project? Please provide more details for the readers to imagine.

Please add the strength and limitation to the discussion.

6. PLOS authors have the option to publish the peer review history of their article (what does this mean?). If published, this will include your full peer review and any attached files.

Reviewer #1: **Yes: **Tahir Khan

Reviewer #2: No

Reviewer #3: No

Reviewer #4: **Yes: **Ramesh Kumar

Reviewer #5: No

---

## [Author Response · Author response to Decision Letter 0]

11 Aug 2020

 Thank you for your comments. We have revised the format according to PLOS ONE style template. 

2. Please address the following:

- Please include additional information regarding the survey or questionnaire used in the study and ensure that you have provided sufficient details that others could replicate the analyses. For instance, if you developed a questionnaire as part of this study and it is not under a copyright more restrictive than CC-BY, please include a copy, in both the original language and English, as Supporting Information.

- Please ensure you have thoroughly discussed any potential limitations of this study within the Discussion section.

Thank you for your comments. We have uploaded our questionnaires as supplementary information and the limitations of the study have been added under Discussion section. (please refer to line 548 – 561 in revised manuscript with track changes)

3.Reviewer comments

Reviewer 1: There are various issues that authors need to rectify in this paper.

1- Was this study representative of the Malaysian population? In my NO, because your study mainly comprised of the Sample from the Melaka state and on top of that the Majority of the respondents are Muslims, so its really challenging to make it compatible with the diverse Malaysian population. However, if the title and methods and other parts where authors have tried to justify this study as a representative of Malaysian population are restricted to Melaka only then it will be more suitable for this study.

Title must have "Melaka" and "Malay Muslims" because your data is more representative to this population instead of the whole Malaysia.

Thank you for your comments. We do agree that this study is not representative of Malaysian students. The study populations were students from public higher educational institutions in Melaka, a state in Malaysia. We have addressed the issue of generalisability of our study as one of our study’s limitation. We also have revised the title of the study. However, we would not specify the title to Malay Muslims because the selection of the study participants was random selection and on voluntary basis. There is a small proportion of non-Malays who volunteered to participate in this study. Furthermore, the students of public of higher educations in Melaka were mainly Malay as Malays are the majority ethnic in Melaka. (please refer to line 411-413 in revised manuscript with track changes)

2- Abstract: Results section, add some more results that give an idea to the readers who were your main study population. Change "Marks" to score.

Thank you for your comments. Additional information has been added under result section in abstract and marks already changed to score (refer to line 56-66 in revised manuscript with track changes). 

3- Introduction: some facts from this report should be in the introduction to give a better idea about the current STI situation in Malaysia https://www.moh.gov.my/moh/resources/Penerbitan/Laporan/Umum/Report_GAM_2019_(Final).pdf

it will be ideal if the following papers are cited as well and the results are compared and contract with them

"Analysis on sex education in schools across Malaysia. Johari Talib, Maharam Mamat, Maznah Ibrahim & Zulkifli Mohamad"

https://www.ncbi.nlm.nih.gov/pmc/articles/PMC5334713/

We appreciate the suggestion given and the references suggested have been included as our references (reference number 7,13,41). (refer line 94, 122, 146, 419, 428, 434, 451, 461 and 494 in revised manuscript with track changes)

4- Methods, Need more explanation about study respondents? also provide details how multistage sampling was performed and how many institutes were approached for consent to this study. For reliability analysis how many subjects were invited for the pilot testing? were these included in the final analysis or excluded. Share the reliability analysis tables with Scale items if deleted options so that readers can see what was the reliability of each item.

Thank you for your comments. We have included more details on study respondents, multistage sampling, and the reliability test under methodology (refer line 161-232 in revised manuscript with track changes)

Standardize throughout MARKS to SCORE

Analysis: in the abstract author claim " Univariate, bivariate and multivariate analyses

were conducted using IBM SPSS software version 25" was performed but in the methods information is missing about this part. need to provide details which variables was included for which analysis what were the co-varaites etc

Thank you for your comments. The details on variable used for each analysis have been provided under “Statistical analysis” section, refer line 264-279 in revised manuscript with track changes)

5- RESULTS

- Reliability and validity additional data need to be provided

- Chi-sq was applied which give association among the variables, I suggest to see the difference among the group not the association. go for t-test and ANOVA to achieve this objective

- Results from the regression model need to be explained in detail and its interpretation need to improve in the results and discussion section

Thank you for your comments. We have provided additional information for reliability and validity of the questionnaires under methodology section. (please refer to line 215- 232). We also have rerun the analysis and applied independent t-test and Mann-Whitney test since the dependent variable used was continuous variable. Multiple linear regression was used in multivariate analysis (please refer to line 264- 279, 369-396 in revised manuscript with track changes).

6- Limitation section is missing

Thank you for your comments. Limitations have been added under discussion) (please refer to line 548-561 in revised manuscript with track changes)

7- Conclusion need revision after re-analysis of the results

Thank you for your comments. Conclusion have been revised accordingly (please refer to line 566-576 in revised manuscript with track changes).

Reviewer 2:

1. Some of the abbreviations were not identified in the context, e.g. AOR.

Thank you for your comments. We have revised all the abbreviations used in the manuscript and provide the full word.

2. Line 117; Reference linked with the sample size estimation; it is not clear how the reference supports the statement.

Thank you for your comments. Since this study involved multistage sampling, the design effect used for sample size calculation was based on the reference linked and sample size was calculated based on previous study (please refer line 169-172 in revised manuscript with track changes) 

3. Line; 150; Is it True that calculated total knowledge score can be considered as continuous variables? Please define how the total knowledge score was calculated.

Thank you for your comments. We have revised the analysis for the study. Knowledge score has been calculated as continuous variable (please refer to line 237 in revised manuscript with track changes)

4. Line 155; Was the constructed logistic regression model adjusted to all variables observed in the study? Please define as appropriate within the context.

Thank you for your comments. We have revised the analysis and conducted the multiple linear regression analyses (please refer to line 264- 279, 369-396 in revised manuscript with track changes).

5. Table 1; different religions seem not to be represented equally in utilised sample, especially "Buddha"? Any explanation?

Thank you for your comments. Majority of the public higher educational institutions in Melaka were reserved for Bumiputera which is Malay ethnicity. Thus, the proportion of other ethnicity and religion were unequally distributed. We already performed Mann Whitney’s U test to accommodate for not normally distributed data in our analysis (please refer to line 269 & 411-413 in revised manuscript with track changes).

6. Table 5; What is the value and role used to dichotomise the total knowledge score into two groups, namely; Adequate and Inadequate? And what is the rationale behind that?

Thank you for your comments. We have revised the analysis and making the dependent variable as continuous data. (Please refer to line 237 in revised manuscript with track changes).

7. Table 2; Revise percentage for " Do screening test", 92.7+7.8=100.5!.

Thank you for your comments. We have revised the calculation and it supposed to be 92.7 + 7.3=100% (please refer to Table 3, line 343 in revised manuscript with track changes)

8. Line 339; please state limitation to this study.

Thank you for your comments. Limitations of the study have been added under the discussion section (please refer to line 548-561 in revised manuscript with track changes).

Reviewer 3

BW

Reviewer #3: Given the current increase in the incidence rates for a variety of sexually transmitted infections in the Asia-Pacific region this paper is timely and relevant. The study design is fairly basic but adequate to address the research questions. There are some issues with how the paper was presented which must be addressed before the paper can be considered for publication.

Introduction

- In Malaysia, the available surveillance systems for STIs reported

an increasing trend of syphilis and gonorrhoea cases, with the incidence rate of syphilis being

only 5.7 per 100,000 population in 2012 but had increased to 8.0 per 100,000 population in

2017 [4, 5]. Similarly, the incidence rate of gonorrhoea was 4.78 per 100,000 population in

2013 but had markedly increased to 10.39 in 2017

These seem to be quite large increases in incidence in malaysia. Is it possible this could also be connected to changes in the surveillance system itself? Is an increase in reporting possible?

Thank you for your comments. A marked increase in the incidence of syphilis and gonorrhoea might be contributed by the improvement in the surveillance systems and the establishment of STI Friendly Clinics in selected government health clinics starting from year 2015 [7]. (please refer to line 91-94 in revised manuscript with track changes).

- The second sentence of the second paragraph starting ' The interactions between their internal conflicts..." does not make sense. I don't know what this sentence is trying to say

Thank you for your comments. We have provided a clearer statement in the paragraph. The sentences were trying to explain the reason for young adults to be more vulnerable to acquire STIs, which we try to relate their psychosexual development and the exposure to favorable situation for risky sexual behaviors such as peer influence and digital media culture (please refer to line 96-101 in revised manuscript with track changes). 

- In the introduction overall, the reader is left a little confused as to what STIs we are interested in. The title and its reference to "Zero city" implies that we will focus on HIV. In the introduction though we mainly hear about syphilis and gonorrhoea. Need to be clear up front what you are interested in analysing.

Thank you for your comments. The study mainly focused on knowledge of STIs and we have revised the title (please refer to line 6-7 in revised manuscript with track changes). 

Methods

- There is a missing step in the methods section between describing the study population and how they were sampled and then the measures section. We need to know how the students were actually recruited and by who. Was the contact through their education institution? Or did the researchers approach them directly? How were the questionnaires actually administered? Online? This research is on quite a sensitive subject. The details of what i have asked for here are important for assessing the results of the research. For example, people may answer an online survey more openly than a face to face interview.

Thank you for your comments. We have provided more information on study population, sampling method and data collection procedure (please refer to line 160-193 in revised manuscript with track changes). 

- To interpret Table 2 we need more specifics about what the actual questions asked were, and what the correct answers were. Don't assume your audience know these things already. For example in Table 2 I assume that there was a question like: " which of the following are STIs" but that is never explicitly said in the methods to this paper. Continuing on i assume that there is a question like: "Which of the following are modes of transmission for STIs" but i don't know for sure.

Thank you for your comments. We have provided the questionnaires form which the reader can refer for further information (S1-Appendix)

Results

- You are missing a table 4. You go straight from 3 to 5.

 Thank you for your comments. We have revised the analysis and new table have been provided (please refer to line 350 in revised manuscript with track changes).

- In Table 5 you have categories of knowledge as 'Adequate' or 'in adequate'. I may have missed it but i can't find definitions for these categories. This should be in the Methods somewhere. Even if it is in the Methods you should include it again as a footnote to the table.

 Thank you for your comments. We have revised the analysis and the knowledge score has been measured as continuous variable (please refer to line 237, 336 in revised manuscript with track changes). 

Discussion

- You observe that your study sample had a higher knowledge score than was found in other studies. I know this could be for a variety of reasons but in the discussion you should think about what these reasons may be. For example was the Thai study in second para of Discussion also amongst university students?

Thank you for your comments. We have provided reason for higher score observed in our study in the discussion. Yes, the study in Thailand was conducted among university students (please refer line 427 in the revised manuscript with track changes).

Reviewer 4

Reviewer #4: Why getting to zero city included in the title? Overall title needs to be revised.

Rationale needs to be included. What we already know about this title is missing.

Thank you for your comment. We have revised the title (please refer to line 6 in revised manuscript with track changes). 

How knowledge was measured? What was the overall Knowledge level? Why three options; Yes, No and DN were included. To see the Knowledge, Yes OR NO can be sufficient. Did you validated / pretested the tool in local context?

Thank you for your comments. We reanalyzed the knowledge as continuous variable. The questionnaires have been validated and tested for its reliability on 60 students from one of the public higher educational institutions in Melaka. Further information on validity and reliability test were provided under methodology section (please refer to line 215-229 in revised manuscript with track changes).

Objectives and title both should be revised as per the results. Author has analyzed the association and predictors of Knowledge as well.

Thank you for your comments. We have revised the title and objectives We have revised the title (please refer to line 6 &152-153 in revised manuscript with track changes). 

Conclusion also needs to be revised.

Thank you for your comments. We have revised the conclusion based on the results of the new analysis (please refer to line 572 in revised manuscript with track changes).

Overall this needs lot of work to be revised.

Thank you for your comments. We have improved the title, methodology section, analysis part, result, and discussion part. 

Reviewer #5: Knowledge on sexually transmitted infections among undergraduate students in

“Getting to Zero City”, Malaysia: A cross-sectional study

Journal: Plos One

Reviewer’s comments

The manuscript by Mansor et al. presents the results of an original study. They collected data about STI knowledge from undergrad students in a large city in Malaysia using questionnaires.

The study conduct, statistic methods use, and other analyses are appropriate.

The reasonable conclusions are presented. To me the language use is understandable.

I have several comments as follows;

Title

-The full title should include the city name “Mekala”

Thank you for your comments. We have changed the title into “Determinants of knowledge on sexually transmitted infections among students in public higher education institutions in Melaka state, Malaysia”. (please refer to line 6 in the manuscript with track changes).

-The short title on the title page

“Predictors of knowledge of sexually transmitted infections among undergraduate students.”

is different from the one in the submission system

“Knowledge on sexually transmitted infections among undergraduate students in Malaysia.”

Please reconcile.

Thank you for your comments. We have removed the short title in the front page after following the manuscript body formatting guidelines and change the short title in the system as “Determinants of knowledge of sexually transmitted infections among undergraduate students”.

Abstract

-Introduction: the first sentence “….is an alarming issue”, please specify that it is an alarming issue of what.

Thank you for your comments. We have rephrased the sentence (please refer to line 42 in revised manuscript with track changes). 

-Results: The response rate for this study was 88%. Please specify whether the authors mean response rate participate in the study (other 12% denied to join) or questionnaire completion rate (12% of questionnaires were not returned, or 12% of questions in the questionnaires were not answered).

12 % refer to those who did not complete the questionnaires (more than 5% of the questionnaires were not complete) and they were excluded in the final analysis. (please refer to line 297-303 in revised manuscript with track changes). 

-Conclusion: what did the authors mean by the term “lower level of education” (all were undergrad student; did they mean the first-year student? It should be clarified)

Thank you for your comments. We have provided more information on the study participants under methodology part. The selected institutions in our study have offered a range of post-secondary courses that range from Malaysia Skill Certificate, diploma, and degree level. Lower level of education refers to those who are not taking degree courses which are Malaysia Skill Certificate and diploma (please refer to line 240-244 in revised manuscript with track changes).

Introduction

In the 4th paragraph, please provide background information about the “Getting to zero” project. Was it implemented in the city or in the university campus. Who were the target population that joined the activities? What kind of thing can people learn from the project?

Thank you for your comments. The project is implemented at city level and main aim of the project is to intensify the efforts towards elimination of HIV/AIDS. The project emphasizes on collaborative effort between different agencies within Melaka including higher educational institutions. Among the objectives of ‘Getting to Zero’ project is to increase the general knowledge and the awareness on HIV and to reduce the high-risk behaviour practices especially among young population. The activities included in the project are surveillance program, screening program, counselling and treatment services, health education and etc. The project primarily aims the key population for HIV (please refer to line 133-145 in revised manuscript with track changes). 

Methodology

-Ethic approval: typo error in the last sentence, it should be “…will be disposed five years after the completion of the study”

Thank you for your comment. We have improved the sentence (please refer to line 293 in revised manuscript with track changes). 

Results

-Firstly, the same question as in the abstract: The response rate for this study was 88%. Please specify whether the authors mean response rate participate in the study (other 12% denied to join) or questionnaire completion rate (12% of questionnaires were not returned, or 12% of questions in the questionnaires were not answered).

Thank you for your comments. We have provided additional information on response rate under result section (please refer to line 297-303 in revised manuscript with track changes). 

-Table 1: Does educational level mean the current or the highest level ever finished?

Thank you for your comments. The current level. (please refer to line 240-244 in revised manuscript with track changes). 

-involvement in STIs program: please clarify in the footnote or somewhere to be referred to what STIs program means.

Thank you for your comments. STIs program refers to any involvement of respondents in any programs or health talk about STIs organized in their college or university or in the community. (please refer to line 249-253 in revised manuscript with track changes). 

-Sociodemographic: Did the authors collect data on sexual behaviours of study participants i.e. gender role, sexual experience, age at sexual debut, sexual partners? Those are interesting and useful information among this population.

Thank you for your comments. No information on sexual behaviour collected for this study

-The 2nd paragraph: line 8, “the top three risk factors that were answer wrongly”. The term should be modified to “the top three risk factors that were misunderstanding”

Thank you for your comments. We have changed the word as suggested (please refer to line 342 in revised manuscript with track changes). 

-Table 5 why the authors used the cut-off age at 23 years?

Thank you for your comments. We rationalized the cut-off 23 years old based on a local study conducted among local university students that have reported those who aged more than 23 years old were reported to have higher knowledge on STIs from the reference below:

Folasayo AT, Oluwasegun AJ, Samsudin S, Saudi SN, Osman M, Hamat RA. Assessing the knowledge level, attitudes, risky behaviors and preventive practices on sexually transmitted diseases among university students as future healthcare providers in the central zone of Malaysia: a cross-sectional study. International journal of environmental research and public health. 2017 Feb;14(2):159.

Some categories are not understandable to general reader, i.e. school vs. not school program.

Thank you for your comments. These categories referring to the type of program involved by the participants. School programs refer to involvement in STIs program in school. Not a school program refers to involvement of STIs in the college and community. We have provided information regarding variables in this study under variable section (please refer to line 251-253 in revised manuscript with track changes). 

In “History of SRH education” there are 2 groups, yes and no.

If it was the standard SRH education in school, why not everyone has ever attended them prior to continue their higher education?

Thank you for your comments. There was no specific curriculum or subject were named as SRH in the school’s curriculum. SRH topics were embedded in other subjects such as biology and family health, Islamic education, moral physical and health education subjects. The question was asked to ensure the participants were aware regarding SRH education delivered in the schools. (please refer to line 253-257 in revised manuscript with track changes). 

Were there any differences in characteristics of student who stay in and out of campus? Was it more expensive to stay outside? Were there any specific inclusion criteria for students who get accommodation in the campus i.e. come from far away town, study in some faculties, or final year student who need to work late at night? Those might be confounders which create difference between those who live in and out of campus.

Thank you for your comments. Unfortunately, no further information collected on the characteristics of the students who live out campus. Basically, we specifically asked the place of stay because those who live out campus usually did not compulsory to join any educational programs held in the campus. While those who lived in the campus are compulsory to join the program. Under the National Strategic Plan Ending AIDS 2016-2030, each of higher educational institution is compulsory to organize an educational program on HIV/STIs once a year. Thus, if the students stay in campus, they might have recently exposed to the information on HIV/STIs and it will influence their knowledge on STIs. (please refer to line 533-540 in revised manuscript with track changes).

Discussion

In the first paragraph, the author mentioned including lecturers and students’ affair department in the study. This would make the study more vulnerable for coercion. Please explain how they involved in the study recruitment or other activities.

Thank you for your comments. We have provided further explanation on data collection procedure under methodology section (please refer to line 173-192 in revised manuscript with track changes).

What were the interventions in the “Getting to zero city” pilot project? Please provide more details for the readers to imagine.

Thank you for your comments. We have provided additional information on the project. (please refer to line 134-145 in revised manuscript with track changes).

Please add the strength and limitation to the discussion.

Thank you for your comments. We have provided the strength and limitation of the study under discussion section (please refer to line 547-567 in revised manuscript with track changes).

---

## [Decision Letter · Decision Letter 1]

23 Sep 2020

PONE-D-20-11879R1

Determinants of knowledge on sexually transmitted infections among students in public higher education institutions in Melaka state, Malaysia

PLOS ONE

Dear Dr. Ahmad,

Thank you for submitting your manuscript to PLOS ONE. After careful consideration, we feel that it has merit but does not fully meet PLOS ONE’s publication criteria as it currently stands. Therefore, we invite you to submit a revised version of the manuscript that addresses the points raised during the review process.

We look forward to receiving your revised manuscript.

Kind regards,

Siyan Yi, MD, MHSc, PhD

Academic Editor

PLOS ONE

Additional Editor Comments:

Thank you for your revisions, addressing all the major concerns raised by the reviewers. As you could find below, reviewer #3 still has some minor comments. Please address these comments carefully. Also, please take this opportunity to improve the writing quality, cleaning the grammatical errors and typos, minimizing passive voice use, and avoiding long and complex sentences. You may not have a chance to proofread your article should the journal accept it for publication.

Reviewers' comments:

Reviewer's Responses to Questions

**Comments to the Author**

1. If the authors have adequately addressed your comments raised in a previous round of review and you feel that this manuscript is now acceptable for publication, you may indicate that here to bypass the “Comments to the Author” section, enter your conflict of interest statement in the “Confidential to Editor” section, and submit your "Accept" recommendation.

Reviewer #1: All comments have been addressed

Reviewer #3: All comments have been addressed

Reviewer #5: All comments have been addressed

2. Is the manuscript technically sound, and do the data support the conclusions?

Reviewer #1: Yes

Reviewer #3: Yes

Reviewer #5: Yes

3. Has the statistical analysis been performed appropriately and rigorously? 

Reviewer #1: Yes

Reviewer #3: Yes

Reviewer #5: Yes

4. Have the authors made all data underlying the findings in their manuscript fully available?

Reviewer #1: Yes

Reviewer #3: Yes

Reviewer #5: Yes

5. Is the manuscript presented in an intelligible fashion and written in standard English?

Reviewer #1: Yes

Reviewer #3: Yes

Reviewer #5: Yes

6. Review Comments to the Author

Reviewer #1: All comments are addressed and no further corrections are required. However, paper may get more clarity upon getting English editing.

Reviewer #3: (No Response)

Reviewer #5: Review manuscript PONE-D-20-11879R1

“Determinants of knowledge on sexually transmitted infections among students in public higher education institutions in Melaka state, Malaysia".

The authors have successfully addressed all reviewers’ comment.

I now have only few minor comments/editing points as follows;

-Page 7 line 166, there should be a period after the close parenthesis.

-Page 7 line 169, the letter “s” should be removed.

-Page 8, the Table 1 should have a bottom line.

-Page 13 There are several unclear terms in the Table 3 that might require clarification.

“having sex man with man” should be either “homosexual relationship” or “same sex relationship”

“Abnormal discharges” should be accompanying with the organ where it came from i.e. vagina, nasal, or urethral.

“Be faithful” should be written in a full term as “Be faithful in the intimate relationship”

-Please specify the full term of abbreviations used in the Tables in each footnote, to make them easier for readers to read and understand at a glance.

7. PLOS authors have the option to publish the peer review history of their article (what does this mean?). If published, this will include your full peer review and any attached files.

Reviewer #1: **Yes: **Tahir M Khan

Reviewer #3: **Yes: **Matthew Kelly

Reviewer #5: **Yes: **Linda Aurpibul MD. MPH.

---

## [Author Response · Author response to Decision Letter 1]

29 Sep 2020

Response to Reviewers.

Reviewer #1: All comments are addressed, and no further corrections are required. However, paper may get more clarity upon getting English editing.

Thank you for your comment: We have sent for English proofread. The amendments are highlighted by the track changes.

Reviewer #5: Review manuscript PONE-D-20-11879R1

“Determinants of knowledge on sexually transmitted infections among students in public higher education institutions in Melaka state, Malaysia".

The authors have successfully addressed all reviewers’ comment.

I now have only few minor comments/editing points as follows;

-Page 7 line 166, there should be a period after the close parenthesis.

Thank you for your comment. We have addressed it. See Page 7, line 172.

-Page 7 line 169, the letter “s” should be removed.

Thank you for your comment. We have addressed it. See Page 8, line 175.

-Page 8, the Table 1 should have a bottom line.

Thank you for your comment. We have addressed it. See Page 8, Table, 1.

-Page 13 There are several unclear terms in the Table 3 that might require clarification.

“having sex man with man” should be either “homosexual relationship” or “same sex relationship”

“Abnormal discharges” should be accompanying with the organ where it came from i.e. vagina, nasal, or urethral.

“Be faithful” should be written in a full term as “Be faithful in the intimate relationship”

Thank you for your comment. We have addressed these. See Page 13, Table 3.

-Please specify the full term of abbreviations used in the Tables in each footnote, to make them easier for readers to read and understand at a glance.

Thank you for your comment. We have addressed these. See Page 8, Table 1. Page 12, Table 2. Page 14, Table 3. Page 14, Table 4. Page 15, Table 5. Page 16, Table 6.

---

## [Editor Report · Decision Letter 2]

5 Oct 2020

Determinants of knowledge on sexually transmitted infections among students in public higher education institutions in Melaka state, Malaysia

PONE-D-20-11879R2

Dear Dr. Ahmad,

We’re pleased to inform you that your manuscript has been judged scientifically suitable for publication and will be formally accepted for publication once it meets all outstanding technical requirements.

Kind regards,

Siyan Yi, MD, MHSc, PhD

Academic Editor

PLOS ONE
---

## [Editor Report · Acceptance letter]

13 Oct 2020

PONE-D-20-11879R2 

Determinants of knowledge on sexually transmitted infections among students in public higher education institutions in Melaka state, Malaysia 

Dear Dr. Ahmad:

I'm pleased to inform you that your manuscript has been deemed suitable for publication in PLOS ONE. Congratulations! Your manuscript is now with our production department. 

Kind regards, 

on behalf of

Dr. Siyan Yi 

Academic Editor

PLOS ONE